# Identification of TLR2 Signalling Mechanisms Which Contribute to Barrett’s and Oesophageal Adenocarcinoma Disease Progression

**DOI:** 10.3390/cancers13092065

**Published:** 2021-04-25

**Authors:** Ewelina Flis, Gillian Barber, Ciara Nulty, Brian Keogh, Peter McGuirk, Akanksha Anand, Jacintha O’Sullivan, Michael Quante, Emma M. Creagh

**Affiliations:** 1School of Biochemistry and Immunology, Trinity Biomedical Science Institute (TBSI), Trinity College Dublin, D02 R590 Dublin, Ireland; flise@tcd.ie (E.F.); barberg@tcd.ie (G.B.); nultyci@tcd.ie (C.N.); keogh.brian@nuritas.com (B.K.); mcguirk.peter@gmail.com (P.M.); 2Department of Internal Medicine, Technical University of Munich, D-80333 Munich, Germany; akanksha.anand@tum.de (A.A.); michael.quante@tum.de (M.Q.); 3Department of Surgery, Trinity Translational Medicine Institute, Trinity Centre for Health Sciences, St. James’s Hospital, D08 W9RT Dublin, Ireland; osullij4@tcd.ie

**Keywords:** oesophageal adenocarcinoma, Barrett’s oesophagus, TLR2 signalling, HMGB1, inflammation

## Abstract

**Simple Summary:**

Oesophageal adenocarcinoma (EAC) is a common type of oesophageal cancer with a rapidly rising incidence. Risk factors such as reflux, smoking, obesity and Barrett’s oesophagus cause chronic irritation and inflammation in the oesophagus. A receptor that causes inflammation, called Toll-like receptor 2 (TLR2), is expressed at higher levels in oesophageal cells from patients with Barrett’s and EAC, compared to disease-free patients. This study aimed to identify mechanisms involved in TLR2-mediated inflammation in oesophageal cells; and to assess whether TLR2 represents a therapeutic target to limit EAC development. Findings reveal that TLR2 activation in Barrett’s organoids and oesophageal cancer cells amplifies inflammation and promotes cancer development by causing the secretion of several inflammatory factors, most notably the nuclear protein, HMGB1. We demonstrate that TLR2 neutralisation efficiently blocks the inflammatory effects of TLR2 in these systems, revealing the therapeutic potential of TLR2 targeting to limit oesophageal disease and cancer progression.

**Abstract:**

Chronic inflammation plays an important role in the pathogenesis of oesophageal adenocarcinoma (EAC) and its only known precursor, Barrett’s oesophagus (BE). Recent studies have shown that oesophageal TLR2 levels increase from normal epithelium towards EAC. TLR2 signalling is therefore likely to be important during EAC development and progression, which requires an inflammatory microenvironment. Here, we show that, in response to TLR2 stimulation, BE organoids and early-stage EAC cells secrete pro-inflammatory cytokines and chemokines which recruit macrophages to the tumour site. Factors secreted from TLR2-stimulated EAC cells are shown to subsequently activate TLR2 on naïve macrophages, priming them for inflammasome activation and inducing their differentiation to an M2/TAM-like phenotype. We identify the endogenous TLR2 ligand, HMGB1, as the factor secreted from EAC cells responsible for the observed TLR2-mediated effects on macrophages. Our results indicate that HMGB1 signalling between EAC cells and macrophages creates an inflammatory tumour microenvironment to facilitate EAC progression. In addition to identifying HMGB1 as a potential target for early-stage EAC treatment, our data suggest that blocking TLR2 signalling represents a mechanism to limit HMGB1 release, inflammatory cell infiltration and inflammation during EAC progression.

## 1. Introduction

Oesophageal cancer represents the sixth most common form of cancer worldwide [1], and its increasing incidence and poor prognosis make it a significant global public health problem. The incidence of EAC is increasing at an alarming rate (>6 fold in the past two decades) in the Western world [2], with gastro-oesophageal reflux disease and obesity representing the major risk factors [3]. It is hypothesised that signalling pathways activated by host–bacterial interactions contribute to the progression of EAC [4] and studies have demonstrated significant differentials in oesophageal microbiome composition between healthy subjects and those with esophagitis or BE [5]. The oesophageal disease–associated microbiome has similar ratios of Gram-negative and -positive bacteria [5], which are likely to engage pathogen recognition receptors, such as Toll-like receptors (TLRs), on epithelial and immune cells to stimulate pro-inflammatory cytokine production and promote persistent oesophageal inflammation. In the oesophagus, long-standing inflammation caused by acid reflux can cause the replacement of squamous cells with columnar epithelial cells (known as Barrett’s metaplasia). Over time, Barrett’s metaplasia may begin to develop features of dysplasia, defined by abnormal epithelial cell development, growth or differentiation [6]. Patients with Barrett’s metaplasia are at increased risk of developing oesophageal adenocarcinoma (EAC), showing >50-fold greater risk of cancer development than the healthy population [7]. Barrett’s dysplasia is histologically classified into low-grade dysplasia (LGD) and high-grade dysplasia (HGD), with HGD considered a pre-malignant condition with an increased risk of developing cancer.

Dysregulated TLR signalling is associated with a number of inflammation-associated malignancies [8]. TLR engagement on cancer cells can stimulate the release of chemokines and danger-associated molecular patterns (DAMPs), which can promote cancer survival, tumour progression and regulate immune responses within the tumour environment [9]. TLR2, in concert with TLR1 or TLR6, triggers inflammatory signalling pathways in response to a range of stimulants, including microbial lipoproteins (diacyl and triacylglycerol moieties) and endogenous activators derived from altered or damaged host tissue or cells [8,9,10,11]. Endogenous TLR2 activators include cardiac myosin [12], hyaluronan fragments [13], heat shock proteins [14] and High Mobility Group Box 1 (HMGB1) [15]. TLR2 overexpression has been observed in Barrett’s and EAC patient tissue [16,17]. Neutralisation of TLR2 is reported to be effective in suppressing TLR2-driven gastric responses and alleviating tumorigenesis in a murine gastric tumour model [18]. We therefore hypothesised that TLR2-mediated signalling contributes to BE and EAC disease progression. Oesophageal cell lines derived from increasing stages of disease progression were used to characterise TLR2 expression and functional responsiveness. Findings show that TLR2 stimulation of dysplastic and early-stage EAC cells leads to TLR2 signalling amplification, via release of the endogenous TLR2 ligand, HMGB1. Tumour-secreted HMGB1 was also shown to activate primary macrophages to a tumour-associated phenotype. We demonstrate that TLR2 blockade reduces the effects of tumour-secreted HMGB1 on macrophages. Findings therefore suggest that that TLR2 represents a promising therapeutic target to limit disease progression to EAC.

## 2. Materials and Methods

### 2.1. Cell Culture and Stimulation

Cell lines were obtained from ECACC (SK-GT 4, OE33, FLO-1); ATCC (GO (CP-B)); and Invivogen (THP1-XBlueTM-CD14 (NF-kB/AP-1- Reporter Monocytes)). Cell lines were maintained in culture for no more than 20 passages. Cells were not authenticated in the past year. Cell lines from ECACC were tested for mycoplasma contamination twice per year using PCR as described by Young et al. [19]. SK-GT 4, OE33, FLO-1 and THP1-XBlue-CD14 were cultured in RPMI 1640 GlutaMAX (Thermo Fisher Scientific, Waltham, MA, USA, Catalog no. 61870036) with 10% foetal bovine serum (Labtech, Sorisole, Italy, Catalog no. FCS-SA/500) and penicillin–streptomycin (100 U/mL, 100 µg/mL, Catalog No. 15140148). GO were cultured in BEGM™ Bronchial Epithelial Cell Growth Medium Bullet kit (Lonza, Basel, Switzerland, Catalog no. CC-3170). Where indicated, cells were pre-incubated (1 h) with the TLR2 neutralising antibody (αTLR2 (0.4–40 μg/mL), T2.5, Invivogen) prior to stimulation. Cells were stimulated using Pam3CSK4 (P3C; 0.05 μg/mL, Invivogen, San Diego, CA, USA, tlrl-pms), Pam2CSK4 (P2C; 0.05 μg/mL, Invivogen, tlrl-pm2s-1), recombinant HMGB1 expressed from HEK 293 (stock conc. 50 µg/mL; Sigma-Aldrich, St. Louis, MO, USA; SRP6265), and LPS from *Escherichia coli* 0111:B4 (1 μg/mL, Sigma-Aldrich, L2630), or with 50% conditioned media (CM), generated as described below.

### 2.2. Preparation of BMDMs

Femurs and tibias of 10–12-week-old C57BL/6 mice were harvested aseptically. Bone marrow cells were flushed out from the bones using sterile 25G needles and a syringe filled with PBS. Bone marrow cells were centrifuged, and cell pellet was resuspended in red blood cell lysis buffer. Isolated cells were seeded on T175 flask and cultured for 7 days in complete DMEM medium (Thermo Fisher Scientific Ltd., Dublin 15, Ireland Catalog no.11965084) supplemented with 20% L929 media.

### 2.3. Murine Organoid Culture

Tissue from the cardia region at the squamous-columnar junction (SCJ) of 12-month old L2-IL-1B mice was processed for 3D organoid culture in Matrigel for 3 weeks, as previously described [20,21]. Three days before harvesting, organoids were pre-treated (1 h) with the αTLR2 neutralising antibody (10 μg/mL) followed by 24 h stimulation with P2C (40 ng/mL). The next day, media was replaced and re-treated with αTLR2 Ab and P2C for a further 24 h. Organoid images were taken under a light microscope (Zeiss, Axiovert 200M), and are representative of organoid morphology at 0, 24 and 48 h.

### 2.4. Conditioned Media

Oesophageal cell lines (GO, SK-GT 4 and OE33) (seeded at 2 × 10^5^ cells/mL; 1.6 × 10^5^ cells/mL; 2 × 10^5^ cells/mL, respectively) were left unstimulated or stimulated with P3C (0.05 μg/mL), P2C (0.05 μg/mL) or LPS (Sigma-Aldrich Ireland Ltd., Co. Wicklow, Ireland, 1 μg/mL) for 4 h. The medium was removed, cells were washed x3 with PBS and incubated in fresh media for 24 h. The resulting supernatant, termed conditioned medium (CM), was used to stimulate THP1-XBlue-CD14 or BMDM at 50% final conc.

### 2.5. QUANTI-BlueTM Assay

THP1-XBlue-CD14 cells were cultured in RPMI (10% FBS, 1% Pen-Strep, G418 (250 μg/mL), Zeocin (200 μg/mL)), seeded at 4x106 cells/mL and cultured o/n prior to stimulation. QUANTI-BlueTM (Alkaline phosphatase detection medium, Invivogen, Toulouse, France) was prepared according to the manufacturer’s recommendations. A volume of 40 μL supernatant was added to 160 μL QUANTI-BlueTM and incubated for 2 h at RT in a 96 well plate prior to absorbance measurement (650 nm).

### 2.6. Detection of Supernatant Proteins

Following stimulation of SK-GT4 cells, supernatants were removed and analysed for HMGB1 by incubation with 1% (*v*/*v*) StrataClean Resin (Agilent Technologies Ireland, Cork, Ireland, 400714) under rotation for 1 h at 4 °C, prior to denaturation of precipitated beads in 1X SDS gel loading buffer. LDH release into supernatants was determined using CytoTox96^®^ Non-Radioactive Cytotoxicity Assay Technical Bulletin (Promega, Mybio Ltd., Kilkenny, Ireland) according to the manufacture’s protocol.

### 2.7. Isolation of Cytoplasmic Proteins

SK-GT4 cells were washed with PBS before resuspension in Subcellular Fractionation Buffer (20 mM HEPES, 10 mM KCl, 2 mM MgCl2, 1 mM EGTA, 1 mM EDTA, 1 mM DTT, supplemented with protease inhibitor cocktail (cOmplete™ ULTRA Tablets, Roche, Merck, Darmstadt, Germany)) and gentle detachment of cells using a cell scraper. Cell suspension was incubated under rotation for 30 min at 4 °C, prior to pelleting the nuclear fraction (centrifugation at 720× *g*, 4 °C, 5 min). The resulting supernatant which contained cytoplasmic protein was collected and stored at 80 °C. Protein concentration was determined with BCA protein assay reagent (Pierce).

### 2.8. Immunoblotting

Whole-cell lysates were generated using RIPA buffer (50 mM Tris, pH 8, 150 mM NaCl, 0.1% (*w*/*v*) SDS, 0.5% (*w*/*v*) sodium deoxycholate and 1% (*v*/*v*) NP-40, supplemented with protease inhibitor cocktail (cOmplete™ ULTRA Tablets, Roche, Basel, Switzerland)). The 10 μg protein samples (whole-cell lysate or cytoplasmic extracts) were run on 12% SDS-PAGE gels, transferred to nitrocellulose membrane and probed with primary antibodies against TLR2 (R&D system; AF2616-SP), caspase-11 (Sigma; C1354, clone 17D9), IL-1β (R&D system; AF-401-NA), NLRP3 (Cell signalling; 15101), HMGB1 (Cell signalling; 3935), Lamin B (Cell signalling, 12586) and actin (Sigma; A3854), followed by incubation with the appropriate HRP-secondary antibody. Immunoblots were developed with enhanced chemiluminescent (ECL) substrate (Millipore, Merck, Darmstadt, Germany) using the BioRad ChemiDoc TM MP Imaging System. Densitometric analysis was performed using Biorad Image Lab software.

### 2.9. Cytokine Measurements

Supernatants were analysed for murine IL-6, IL-10, TNFα, CXCL1 and MIP-2, and human IL-8 and IL-6, using ELISA kits (Biolegend, San Diego, CA, USA; R&D System) according to the manufacturer’s protocols.

### 2.10. Quantitative Real-Time PCR

Gene expression was analysed using SYBRTM Green PCR Core Reagents (Invitrogen) using Step One Plus, v2.3 (Applied Biosystem). TLR2 mRNA was detected using forward (ATCCTCCAATCAGGCTTCTCT) and reverse (ACACCTCTGTAGGTCACTGTTG) primers. Results were normalised to the housekeeping gene GAPDH.

### 2.11. Statistical Analysis

Data are presented as the mean ± SEM. Statistical significance was analysed using the GraphPad Prism 5.0 statistical program (GraphPad Software, La Jolla, CA, USA). Data sets were analysed using the parametric Student’s *t*-test to compare treated cells to untreated cells or ANOVA for grouped analysis. Data were considered significant when * *p* < 0.05; ** *p* < 0.01; *** *p* < 0.001.

## 3. Results

### 3.1. Cell Lines Derived from Barrett’s Dysplasia and Early Oesophageal Adenocarcinoma Are Responsive to TLR2

To characterise TLR2 expression during oesophageal adenocarcinoma (EAC) disease progression, cell lines derived from Barrett’s high-grade dysplasia (pre-cancerous GO cell line) and increasing stages of distal EAC were examined. The EAC cell lines used were SK-GT 4 (derived from well-differentiated, stage IIB EAC), OE33 (derived from poorly-differentiated, stage IIA EAC) and FLO-1 (derived from poorly-differentiated, stage III EAC). The basal expression of TLR2 mRNA was detectable by RT-PCR in the dysplastic and EAC cell lines, with the exception of FLO-1 (Figure 1a). The highest TLR2 mRNA levels were observed in OE33 cells (Figure 1a). To assess TLR2 protein levels in response to TLR2 stimulation, cell lines were stimulated with the synthetic TLR2 ligands, Pam3CSK4 (P3C; triacylated lipopeptide) and Pam2CSK4 (P2C; diacylated lipopeptide), which activate TLR1/2 and TLR2/6, respectively. Although basal TLR2 expression levels are low in GO and SK-GT 4 cells, Western blot reveals that stimulation with TLR2 ligands causes its upregulation in GO, SK-GT 4 and OE33 cells (Figure 1b). Similar to TLR2 mRNA observations, no TLR2 protein expression was detected in FLO-1 cells (Figure 1b). Secretion of IL-6 and IL-8 in TLR2-responsive oesophageal cells was analysed following their stimulation with a range of TLR agonists. Results reveal that GO cells secrete the highest concentrations of IL-6 in response to TLR2 stimulation (Figure 1c). The EAC cell lines SK-GT 4 and OE33 secrete significant levels of IL-8 in response to TLR stimulation, while IL-6 secretion was low (Figure 1d,e). Chemokine production is an essential factor in inflammation-mediated tumour progression, and IL-8 signalling mediates the invasiveness of EAC-derived cells [22]. Overall, these observations suggest that Barrett’s dysplasia and early-stage EAC cells have pro-inflammatory responses to TLR2 stimulation, which may be important during cancer progression.

### 3.2. Oesophageal Cell Responses to TLR2 Stimulation Are Inhibited by a TLR2 Neutralising Antibody

To assess whether the TLR2 antagonistic antibody (T2.5, αTLR2) inhibits TLR2-induced IL-8 secretion, GO (Figure 2a,b) and SK-GT 4 (Figure 2c,d) cells were pre-incubated with αTLR2 prior to stimulation with the TLR2 ligands, P2C (Figure 2a,c) or P3C (Figure 2b,d). αTLR2 significantly inhibited IL-8 release at 10 μg/mL in the BE cell line (GO) and at all concentrations tested in the early-stage EAC cell line, SK-GT 4 (Figure 2a–d). The effect of αTLR2 on autocrine TLR2 upregulation was determined by Western blot, which shows that αTLR2 pre-treatment of GO and SK-GT 4 cells inhibits TLR2 upregulation (Figure 2e). These results confirm that TLR2 neutralisation effectively inhibits functional responses to TLR2 stimulation in cell lines derived from dysplastic and early-stage EAC.

### 3.3. TLR2 Inhibition Limits Chemokine Secretion from Oesophageal Organoids

Organoids are defined as three-dimensional, self-organising, stem cell-derived structures that resemble their in vivo tissue counterparts [23]. The organoid model system represents a pre-clinical tool which can provide a better mechanistic understanding of Barrett’s and EAC progression [24] and has enormous potential in the evaluation of drug efficacy and toxicity [25]. We wanted to confirm the efficacy of αTLR2 in diseased oesophageal organoids. The transgenic L2-IL-1B mouse is an established model for Barrett’s oesophagus [26]. TLR2 upregulation and increased IL-8/KC chemokines have been observed in the oesophagus of these mice during disease progression [27]. Oesophageal organoids were established from crypts isolated from the cardia region at the squamous-columnar junction of 12-month old L2-1L-1B mice, which have well-established metaplasia and initial dysplasia [26]. Murine Barrett’s organoids were pre-treated with αTLR2 prior to P2C stimulation. Although mice do not express a direct IL-8 homolog, CXCL-1 (KC) and CXCL-2 (MIP-2) are the major mediators of leucocyte recruitment in murine tissues [28,29]. Figure 3a shows representative images of organoids at 0, 24 and 48 h, exhibiting growth and budding which is characteristic of crypts cultured in this organoid system [21]. After 48 h organoids were harvested and secretion of CXCL-1 and MIP-2 were assessed in the supernatants (Figure 3b,c). Following TLR2 stimulation, secretion of CXCL-1 and MIP-2 from Barrett’s organoids was significantly upregulated (Figure 3b,c). TLR2 inhibition reduced CXCL1 secretion and completely impaired MIP-2 secretion, suggesting that MIP-2 is directly regulated via TLR2 during Barrett’s (Figure 3b,c).

### 3.4. Secretion of TLR2 Agonists by Oesophageal Cell Lines

Release of endogenous TLR2 ligands from oesophageal disease cell lines may contribute to the increased TLR2 expression that is observed during EAC development [16]. We employed a THP1-XBlue-CD14 reporter cell line which stably expresses an NF-κB- and AP-1-inducible secreted embryonic alkaline phosphatase (SEAP) reporter gene. The cells secrete SEAP following TLR engagement, which can subsequently be spectrophotometrically detected. Direct stimulation of THP1-XBlue-CD14 cells with a range of TLR agonists was initially performed to confirm the validity of the reporter system (Appendix A). Results show that the undifferentiated THP-1-XBlue-CD14 cell line is highly responsive to heat-killed bacteria, unpurified LPS (Sigma), and the TLR2 ligands, P2C and P3C. Furthermore, TLR-mediated activation can be successfully inhibited by αTLR2 (Appendix A).

To determine whether TLR2-activating factors are secreted from stimulated oesophageal cell lines, GO, SK-GT 4 and OE33 cells were left untreated or pre-treated with αTLR2 prior to their stimulation with a range of TLR agonists. Following 4 h TLR stimulation, the medium was removed and cells were washed thoroughly to remove all traces of the TLR activating ligands. Fresh medium was subsequently incubated with cells for 24 h to generate conditioned medium (CM). CM was incubated for 24 h with THP1-XBlue-CD14 cells (50% final volume) before measuring SEAP activity (Appendix A, Experimental Scheme). Results show that CM from TLR2-stimulated oesophageal cell lines induced the activation of TLRs in THP1-XBlue-CD14 cells (Figure 4a–c). Secretion of TLR activating factors was inhibited when oesophageal cells were pre-treated with αTLR2 (Figure 4a–c). This result suggests that TLR2 engagement on dysplastic oesophageal and EAC cell lines induces the secretion of endogenous TLR2 ligands. In contrast, no THP1-XBlue-CD14 cell responses were observed following incubation with CM from oesophageal cell lines stimulated with the TLR4 agonist, LPS, providing further confirmation that the secretion of pro-inflammatory factors from diseased oesophageal cells is TLR2 dependent (Figure 4a–c). To confirm that endogenous TLR ligands were being secreted, and rule out the possibility that synthetic ligands remaining in the CM may be responsible for the activation of THP1-XBlue-CD14 cells, a time-course (1 min–24 h) to generate CM at different incubation times was performed, and the resulting CM was added to THP1-XBlue-CD14 cells before measuring SEAP activity (Figure 4d–f). Results confirm that CM collected directly after synthetic ligands were removed (1 min timepoint) was unable to activate SEAP in THP1-XBlue-CD14 cells (Figure 4d–f). Results also show that secretion of TLR ligands from oesophageal cell lines increased over initial timepoints, with the highest SEAP levels being observed following 4 and 24 h incubation (Figure 4d–f). To determine the extent of TLR2-specific endogenous ligands being secreted from oesophageal cells, THP1-XBlue-CD14 cells were pre-treated with a concentration range of αTLR2 before incubation with CM from the GO, OE33 and SK-GT 4 cell lines. Results reveal that even low αTLR2 concentrations (0.4 μg/mL) inhibit SEAP release from THP1-XBlue-CD14 cells, suggesting that TLR2-activating factors within the CM are fully responsible for this signal (Figure 4g).

### 3.5. Tumour Cells Release HMGB1 Following TLR2 Stimulation

We next sought to determine the oesophageal cell-derived factors responsible for the activation of TLR2 in THP1-XBlue-CD14 cells. We observed that heat inactivation of CM completely inhibited its ability to activate TLRs, suggesting that secreted peptide/protein factor(s) were responsible for TLR2 activation (Appendix A). HMGB1 is one of several ‘alarmins’ that have been reported to signal via TLR2 [30]. A recent study has proposed HMGB1 as a novel biomarker for risk stratification in Barrett’s-associated EAC progression [31]. We therefore analysed HMGB1 expression following TLR2 stimulation of the EAC cell line, SK-GT 4. TLR2 stimulation caused increased expression levels of both TLR2 and HMGB1 in SK-GT 4 cells over 24 h, which were inhibited in the presence of the TLR2 neutralising antibody (Figure 5a). In contrast, the expression of the Receptor for Advanced Glycation End products (RAGE) was not inhibited by the presence of the TLR2 neutralising antibody (Figure 5a). To determine whether translocation from the nucleus and cellular release of HMGB1 occurs following TLR2 stimulation, HMGB1 levels were measured in cytoplasmic extracts and CM from stimulated SK-GT 4 cells over time. Results show that HMGB1 translocation into the cytoplasm occurs within 4 h (Figure 5b,c). Cellular release of HMGB1 into supernatants is detectable at 20 h following P2C stimulation, corresponding with decreased levels of HMGB1 in the cytoplasm (Figure 5b,d).

### 3.6. TLR2-Stimulated EAC Cells Activate Macrophages through HMGB1.

Macrophages play an important role in coordinating inflammation and are known to stimulate key steps in tumour progression [32]. Infiltration of peripheral monocytes into the tumour microenvironment induces their differentiation into tissue macrophages, which can be classified into a range of phenotypes which include pro-inflammatory, classically activated M1 macrophages and anti-inflammatory M2 macrophages, at the two extremes [33]. In addition to the production of pro-inflammatory mediators such as TNFα, IL-1β or IL-6, macrophages activated by TLRs also produce anti-inflammatory cytokines such as IL-10, polarising them into an immunoregulatory M2b phenotype, which drives Th2 responses [34]. To determine the effect of the oesophageal tumour microenvironment on primary macrophages, murine bone marrow-derived macrophages (BMDMs) were incubated with CM from unstimulated and TLR2-stimulated SK-GT 4 cells, prepared in the same manner as for THP1-XBlue-CD14 stimulations (Figure 4). Analysis of BMDM supernatants reveals that CM from both P2C- and P3C-stimulated SK-GT 4 cells induces the secretion of IL-6, TNFα and IL-10 (Figure 6a), suggesting that macrophages are being polarised into an M2b/TAM-like phenotype. Secretion of IL-6, TNFα and IL-10 are inhibited by αTLR2, confirming that ligands present in EAC CM activate macrophages specifically through TLR2 (Figure 6a).

Inflammasome-mediated inflammation is emerging as a mechanism involved in the initiation and progression of cancer, with low-grade inflammation sustained by inflammasome signalling contributing to all stages of tumourigenesis. To determine whether the oesophageal tumour microenvironment can, in addition to its ability to stimulate the secretion of TAM-associated cytokines, stimulate the expression of inflammasome-associated proteins in primary macrophages, lysates from BMDM incubated with CM from SK-GT4 cells were analysed by Western blot. Results reveal that SK-GT 4 CM induces the upregulation of the inflammasome sensor protein, NLRP3, the inducible inflammasome-associated enzyme, caspase-11 and the inducible inflammasome-associated cytokine, pro-IL-1β (Figure 6b). The expression of these inflammasome-associated markers are also governed by TLR2 signalling, as indicated by their significant impairment in the presence of αTLR2 (Figure 6b).

Having observed that TLR2-activated EAC cells release HMGB1 (Figure 5), we wanted to determine its ability to prime primary murine macrophages for inflammasome activation. Results show that recombinant HMGB1 (rHMGB1) is capable of priming BMDM for inflammasome activation, as determined by the upregulation of inflammasome-related proteins, caspase-11 and pro-IL-1β (Figure 6c). The ability of the TLR2 neutralising antibody to inhibit this effect confirms that HMGB1 signalling is occurring via TLR2 (Figure 6c).

To summarise, data generated during this study identify TLR2 signalling as being most relevant during BE and early-stage EAC development, the most inflammatory stages of EAC disease. TLR2-stimulated BE and EAC cells respond to TLR2 agonists by secreting chemokines and tumour-secretory factors which have TLR2 signalling capability. Tumour secreted factors, including HMGB1, can prime macrophages for inflammasome activation and polarise them into a pro-tumourigenic M2-like phenotype via TLR2, suggesting that TLR2 signalling is critically important for the inflammatory and polarising effects of the tumour microenvironment and that TLR2 represents a promising target to limit EAC disease progression.

## 4. Discussion

TLR2 plays a key role in the activation of inflammatory responses via its ability to recognise pathogen and danger-associated signals [35]. Data generated during this study suggest that TLR2 signalling is highly relevant during dysplastic and early stages of EAC disease, whereas TLR2 expression and signalling appear to be downregulated during later stages of malignancy. In support of these findings, a recent report by Huhta and colleagues showed, using immunohistochemistry, that TLR1, 2 and 6 expression levels increased during Barrett’s metaplasia and dysplasia [16]. However, they did not observe significant increases in TLR2 expression in the subsequent progression to adenocarcinoma, with TLR1 and TLR6 showing decreased expression levels in carcinomas when compared with premalignant epithelium [16].

TLR2 is activated by lipoproteins/peptidoglycans from Gram-positive and -negative bacteria and endogenous ligands released during tissue stress [36,37]. Activation of TLR2 leads to downstream NF-κB activated inflammatory signalling. Although no active NF-κB is detectable in the normal oesophagus, high levels of active NF-κB have been detected in BE and EAC tissue [37]. We show that TLR2 expression is upregulated on GO and SK-GT 4 cells following TLR2 stimulation, and that the TLR2 neutralising antibody, αTLR2, can block both TLR2 upregulation and TLR2-mediated IL-8 production. IL-8 expression in oesophageal tissue has been shown to correlate with histopathologic inflammation in BE patients and is strongly expressed in EAC patients, suggesting a role for IL-8 in oesophageal carcinogenesis [38]. It has been shown that increased circulating proinflammatory cytokines, including IL-8, participate in BE development [39].

In support of clinical findings regarding TLR2 expression during BE and early-stage EAC, high-fat diet (HFD)-fed L2-IL-1B mice (a transgenic BE model), also observed increased TLR2 expression during disease progression [27]. Our data show that TLR2 stimulated BE organoids had increased production of CXCL1 (KC) and CXCL2 (MIP-2) chemokines, which were significantly inhibited in the presence of the neutralising antibody, αTLR2. Both CXCL1 and CXCL2 control neutrophil recruitment [40]. Furthermore, increased CXCL1 levels promote oesophageal dysplasia in HFD-fed L2-IL-1B mice, which develop dysplasia more rapidly than mice fed on a normal diet [27]. Results therefore suggest that αTLR2 represents a potential therapeutic to limit TLR2 upregulation and NF-κB-mediated inflammation during BE and progression to EAC. Interestingly, germ-free L2-IL-1B mice show a marked reduction in inflammation and neutrophil influx [27], leading us to speculate that the gut microbiota activates TLR signalling, leading to IL-8 activation during BE disease progression. We hypothesise that since TLR2-activated oesophageal cells produce chemokines, which promote leukocyte trafficking to the tumour microenvironment, they also induce tumour progression via macrophage differentiation.

Monocytes and macrophages play a very important role in tissue homeostasis and immunity. It is well known that monocytes are recruited from circulation in response to chemoattractants released by cancer cells [41]. For example, Lewis lung carcinoma (LLC) cells secrete factors which activate TLR2 in murine macrophages, mediating IL-6 and TNFα production and inducing lung inflammation [42]. We show that factors secreted from TLR2-activated EAC cells can prime macrophages for inflammasome activation and induce macrophage production of cytokines which have characteristics of both M1 and M2 subsets—IL-6, TNFα and IL-10. Several studies indicate that factors secreted from tumour cells can influence differentiation into a M2/tumour-associated macrophage (TAM)-like phenotype [43,44]. Consistent with these findings, elevated M2/M1 ratios in EAC patient resection tissues correlate with lymph node metastasis and poor patient survival rates [45]. Cao et al. demonstrated that co-culture of SK-GT 4 cells with THP-1 monocytes resulted in increased IL-10 and polarisation into M2d-like macrophages, stimulating EAC cell migration and increased cell invasion [45]. Autocrine IL-6 production by macrophages can skew them towards an M2d subtype [46]. Our results therefore suggest that TLR2-stimulated EAC cells secrete factors which can differentiate macrophages into M2/TAM-like macrophages to promote cancer progression. As the effects of EAC cells are blocked by αTLR2, it confirms that endogenous TLR2 ligands represent the secreted factors that are responsible for these effects.

HMGB1 normally functions as a nuclear protein, but more recently it has been reported to be released from cells to function as a cytokine. Many studies are focused on the role of HMGB1 in inflammation and tumour progression [47]. Irradiated pancreatic cancer cells secrete HMGB1 and engage TLR2 to significantly promote cell migration [42]. Exosomal HMGB1 obtained from oesophageal squamous cell carcinoma (ESSC), similar to recombinant HMGB1, can differentiate monocytes into TAM [48]. However, until this study, the effect of HMGB1 in EAC was unclear. We demonstrate that upon TLR2 stimulation, EAC cancer cells upregulate HMGB1, which translocates to the cytosol before being released extracellularly. As blocking TLR2 signalling causes decreased HMGB1 expression in SK-GT 4, combined with our observation that αTLR2 pre-treatment inhibits the ability of EAC CM to activate TLRs on monocytic cells, we hypothesise that HMGB1 is the main factor secreted from EAC cells responsible for TLR2 activation on monocytic cells. Our results also show that exogenous HMGB1, similar to secreted factors, upregulates caspase-11 and pro-IL-1β through TLR2, again confirming the ability of HMGB1 to activate macrophages. Future studies will determine whether, in addition to HMGB1, oesophageal tumour cells also secrete factors including extracellular matrix components (eg. versican, biglycan, heparan sulphate) and heat shock proteins, which have been previously shown to be secreted from tumour cells to activate macrophages via TLR2-induced inflammation [49,50]. Future studies designed to immunodeplete HMBG1 from EAC CM will confirm whether HMGB1 represents the major TLR2-activating component secreted by EAC cells.

Once TLR2 signalling is initiated during BE and early-stage adenocarcinoma, the positive amplification of TLR2-mediated inflammation and TLR2 expression on macrophages contributes to their polarisation and activation during disease progression. We have shown that TLR2 neutralisation blocks the ability of EAC cells to secrete TLR2-activating factors and represents a promising target to limit disease progression in BE and early-stage EAC patients.

## Figures and Tables

**Figure 1 cancers-13-02065-f001:**
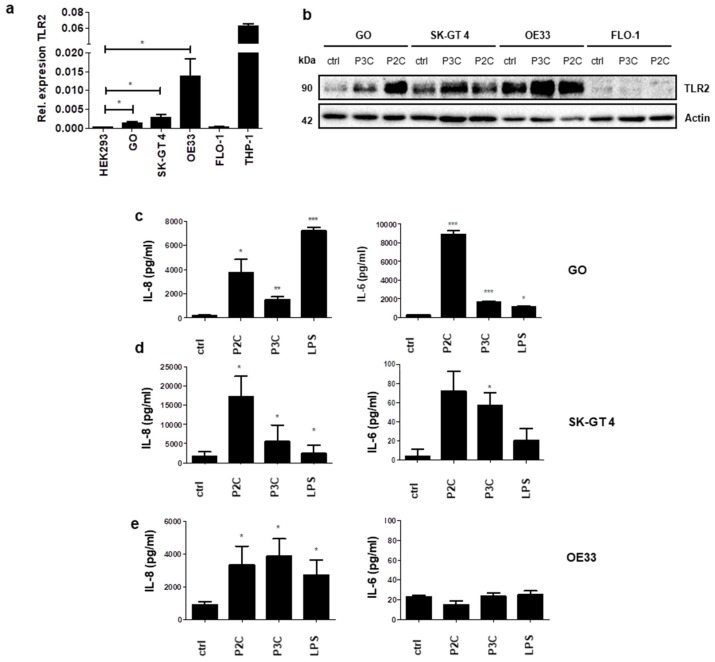
Characterising TLR2 expression and response to TLR2 ligands in a panel of oesophageal cell lines. (**a**) TLR2 mRNA levels were determined in oesophageal cell lines GO, SK-GT 4, OE33 and FLO-1. HEK293 (human embryonic kidney) and THP-1 (human monocytic) cells were included as negative and positive controls, respectively. Data represent mean ± SEM of *n* = 3; * *p* < 0.05 (unpaired two-tailed Student’s *t*-test). (**b**) Oesophageal cell lines were left unstimulated (ctrl), stimulated with Pam3CSK4 (P3C, 0.05 μg/mL) or Pam2CSK4 (P2C, 0.05 μg/mL) for 24 h before analysis of lysates for TLR2 expression by Western blot. Blot is representative of three independent experiments. Oesophageal cell lines: (**c**) GO; (**d**) SK-GT 4; and (**e**) OE33; were stimulated for 24 h with: P3C (0.05 μg/mL), P2C (0.05 μg/mL) or LPS (1 μg/mL). IL-6 and IL-8 secretion levels were determined by ELISA. Data shown are representative of three independent experiments. Unpaired two-tailed Student’s *t*-test to compare the mean ± SEM values between treated and untreated cells; * *p* < 0.05; ** *p* < 0.01; *** *p* < 0.001.

**Figure 2 cancers-13-02065-f002:**
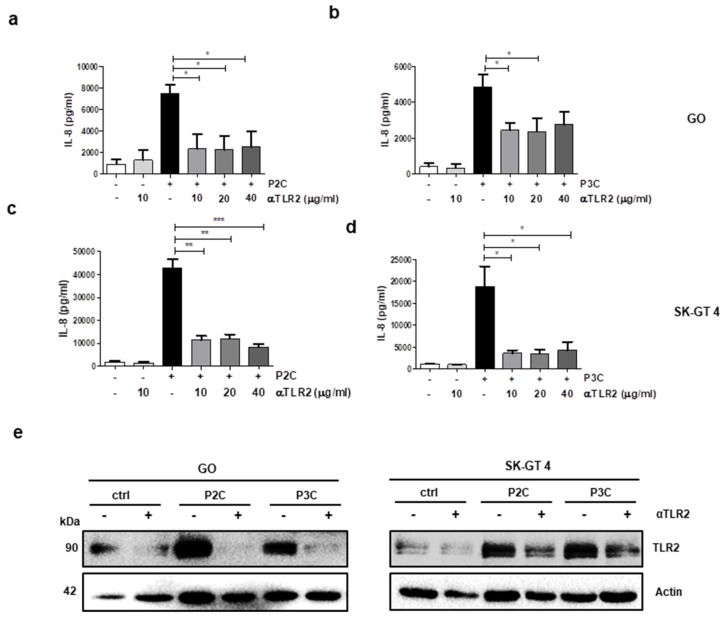
A TLR2 neutralising antibody limits TLR2-mediated IL-8 secretion and TLR2 upregulation in oesophageal cell lines. (**a**,**b**) GO and (**c**,**d**) SK-GT 4 cells were left untreated or pre-treated (1 h) with αTLR2 Ab (10 μg/mL; 20 μg/mL; 40 μg/mL) and subsequently stimulated with P3C (0.05 μg/mL) (**a**,**c**) or P2C (0.05 μg/mL) (**b**,**d**) for 24 h. Supernatants were analysed for IL-8 by ELISA. Data represent the mean ± SEM of *n* = 3; * *p* < 0.05; ** *p* < 0.01, *** *p* < 0.001 (unpaired two-tailed Student’s *t*-test). (**e**) TLR2 expression levels were analysed in untreated or αTLR2 (10 μg/mL) pre-treated (1 h) GO and SK-GT 4 cells, subsequently stimulated with TLR2 ligands, P3C or P2C (each at 0.05 μg/mL). Western blots are representative of three independent experiments.

**Figure 3 cancers-13-02065-f003:**
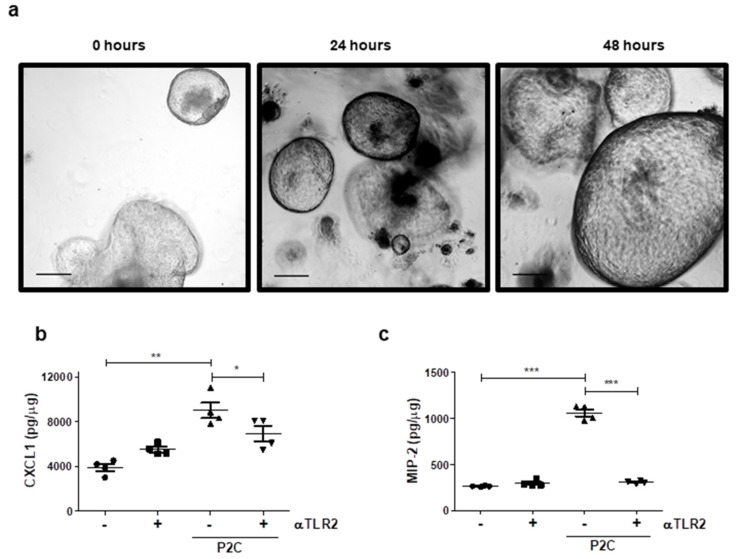
Inhibition of TLR2 in oesophageal organoids from the murine Barrett’s model, pL2-IL-1β, reduces secretion of the IL-8 orthologs, CXCL-1 and MIP-2. Cells taken from the cardia region at the squamous-columnar junction (SCJ) of 12-month old pL2-IL-1β mice (*n* = 4) were cultured (3 weeks) as 3D organoids. Two days prior to harvesting, organoids were pre-treated (1 h) with αTLR2 Ab (10 µg/mL) before stimulation with P2C (40 ng/mL) for 24 h, the medium was replaced and re-treated (αTLR2 Ab followed by P2C stimulation) for a further 24 h. (**a**) Representative images of organoid morphology at 0, 24 and 48 h (magnification: 5×, scale bars—200 µm). Organoids were harvested and supernatants were analysed for (**b**) CXCL-1; and (**c**) MIP-2 secretion. Data were normalised to total organoid protein (pg/µg) and are represented as ± SEM. Paired Student’s *t*-test comparing inhibitor treated to untreated, and unstimulated to stimulated: * *p* < 0.05, ** *p* < 0 01, *** *p* < 0.001.

**Figure 4 cancers-13-02065-f004:**
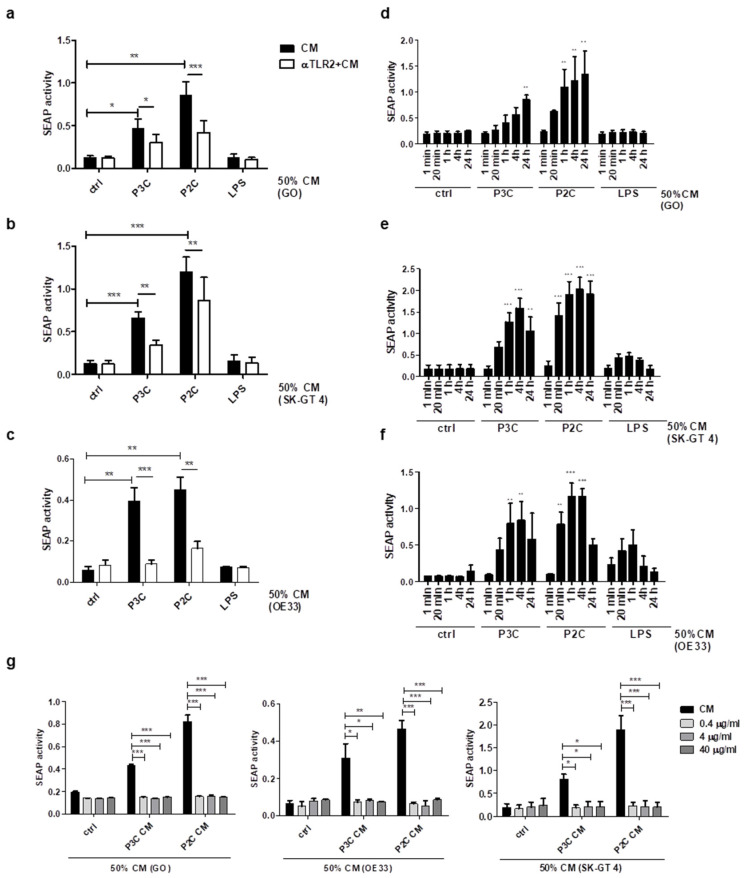
Oesophageal cell lines secrete TLR agonists following TLR2 stimulation. (**a**–**c**) Oesophageal cells (GO, SK-GT 4 and OE33) were αTLR2 (10 μg/mL) pre-treated for 1 h prior to stimulation (4 h) with P3C (0.05 μg/mL), P2C (0.05 μg/mL) or LPS (1 μg/mL). The medium was removed, cells were washed x3 with PBS and incubated in fresh media for 24 h. Conditioned medium (CM) was collected from (**a**) GO; (**b**) SK-GT 4; and (**c**) OE33 cells and added (50% final conc.) to the TLR reporter cell line THP1-XBlue-CD14 (seeded at 4 × 10^6^ cells/mL) for 24 h. TLR stimulation was assessed by measuring supernatant SEAP levels, using QUANTI-Blue reagent. Statistical analysis was performed using an unpaired two-tailed Student’s *t*-test to compare the mean ± SEM values between untreated and CM treated cells and two-way ANOVA to compare αTLR2-treated and -untreated cells, *n* = 3, * *p* < 0.05; ** *p* < 0.01; *** *p* < 0.001. (**d**–**f**) Oesophageal cells were stimulated as in (**a**–**c**) for 4 h. The medium was removed, cells were washed (x3) and incubated in fresh media for the timepoints indicated. CM was collected from (**d**) GO; (**e**) SK-GT 4; (**f**) OE33; and added (50% final conc.) to THP1-XBlue-CD14 cells. Values represent the mean ± S.E.M of three independent experiments; * *p* < 0.05; ** *p* < 0.01; *** *p* < 0.001 (Unpaired Student’s *t*-test). (**g**) THP1-XBlue-CD14 cells were left untreated or pre-treated (1 h) with 0.4 μg/mL; 4 μg/mL; 40 μg/mL αTLR2 prior to incubation with 50% CM collected from unstimulated, P3C- or P2C-stimulated GO, SK-GT 4 or OE33 cells. Data represent the mean ±SEM of *n* = 3; ** *p*  <  0.01; *** *p*  <  0.001 (two-way ANOVA test).

**Figure 5 cancers-13-02065-f005:**
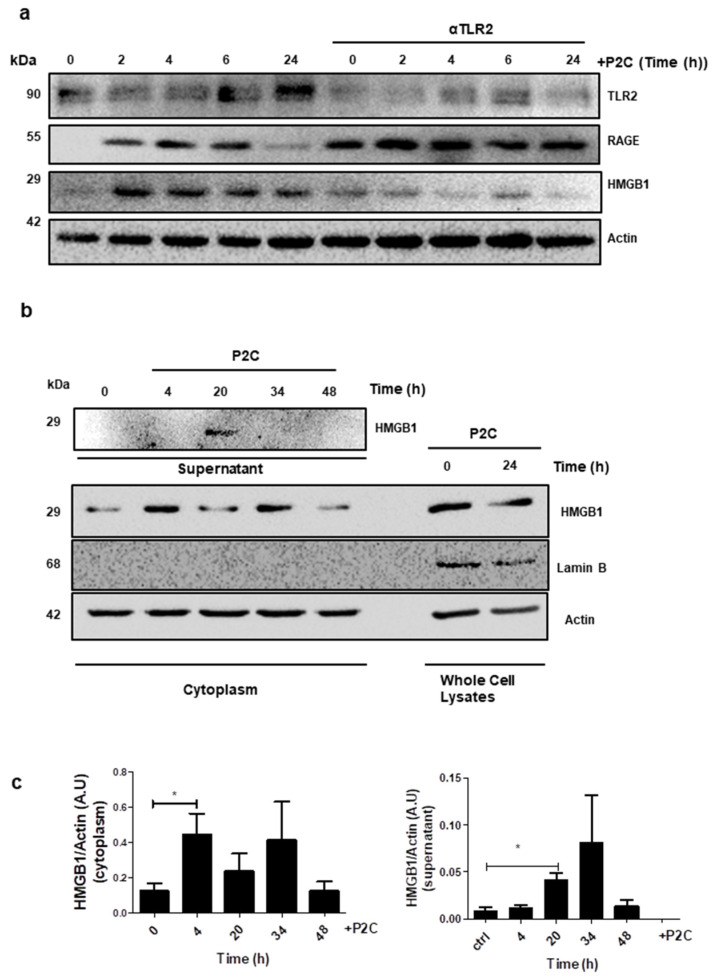
TLR2 stimulation of oesophageal cancer cells induces HMGB1 secretion. SK-GT 4 cells were stimulated with P2C (0.05 μg/mL) for the indicated times. (**a**) Cellular expression levels of TLR2, RAGE, HMGB1 and actin (loading control) were analysed by Western blot. (**b**) HMGB1 levels in supernatants (upper blot) and in the cytoplasm (lower blots) were analysed by Western blot. Whole-cell lysates were used as a positive control for cytoplasmic isolation; Lamin B—nuclear marker; actin—cytoplasmic marker. (**c**) Densitometry of HMGB1 levels in the cytoplasm and supernatants based on blots from three independent experiments. Values represent the mean ± SEM, * *p* < 0.05 (unpaired two-tailed Student’s *t*-test).

**Figure 6 cancers-13-02065-f006:**
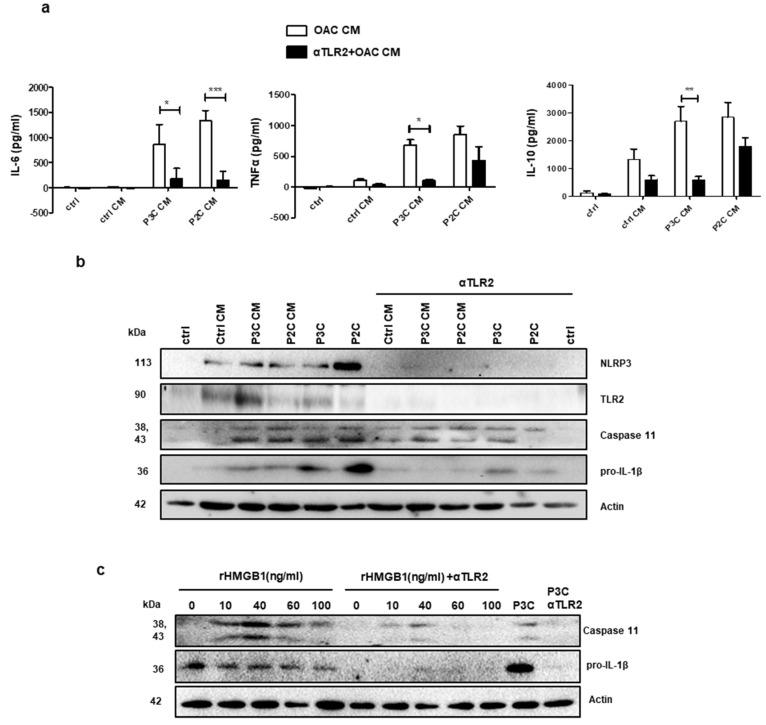
Conditioned media from oesophageal cancer cells induce TLR2-dependent cytokine secretion and upregulation of inflammasome-related proteins in macrophages. (**a**) IL-6, TNFα and IL-10 levels in BMDM supernatants pre-treated with ctrl, P3C- and P2C-CM were determined by ELISA. Values represent the mean ± SEM of *n* = 3; * *p* < 0.05; ** *p* < 0.01; *** *p* < 0.001 (two-way ANOVA test). (**b**) BMDMs were left untreated or pre-treated with αTLR2 (10 μg/mL) for 1 h prior to stimulation (24 h) with 50% CM collected from unstimulated (ctrl CM), P3C- or P2C-stimulated (0.05 μg/mL) SK-GT 4 cells; or directly with P3C or P2C (0.05 μg/mL) as positive control. The expression of NLRP3, TLR2, caspase-11, pro-IL-1β and actin (loading control) were determined by Western blot. (**c**) BMDMs were left untreated or pre-treated with αTLR2 (1 h, 10 μg/mL) prior to stimulation (24 h) with recombinant HMGB1 (10–100 ng/mL). Caspase-11, pro-IL-1β and actin (loading control) levels were determined by Western Blot. Blots are representative of three independent experiments.

## Data Availability

No new data were created or analyzed in this study. Data sharing is not applicable to this article.

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
