# Peer review of "Identification of TLR2 Signalling Mechanisms Which Contribute to Barrett’s and Oesophageal Adenocarcinoma Disease Progression"

_cancers, 2021, doi:10.3390/cancers13092065_

Round 1

Reviewer 1 Report

Improved discussion and methods

Author Response

We thank Reviewer#1 for reassessing the manuscript, and for acknowledging the improved discussion and methods sections.

Reviewer 2 Report

After the review, I believe that the paper has become even more attractive and it will gather more citations in my opinion.

Small final note: 2.1. penultimate line, please write ”Escherichia coli” instead of ”E. coli”, and also please check supplementary figures. You have two supplementary figures 2.

Even though small notes, I strongly recommend publishing the paper. 

Author Response

We thank Reviewer#2 for pointing out these errors in the revised manuscript – we have replaced (E. coli with ”Escherichia coli” in the materials & methods section 2.1 (pg 3, highlighted in blue), and we have corrected the labelling of the Supplementary Figures.

Reviewer 3 Report

This is an interesting manuscript, as the authors identified a mechanisms involved in TLR2-mediated inflammation in esophageal cells.  The authors find that  the endogenous TLR2 ligand, HMGB1, is the factor secreted from EAC cells responsible for the observed TLR2-mediated effects on macrophages. Finally, authors suggest that blocking TLR2 signalling represents a mechanism to limit HMGB1 release, inflammatory cell infiltration and inflammation during EAC progression. These results can be important to increase our knowledge about molecular mechanisms of esophageal adenocarcinoma. However, some issues should be addressed.

  1. My main concern is that at this point the authors do not know whether or not HMBG1 represents the major TLR2 activating component secreted by EAC cells.
  2. Why the authors used 10-12 week-old C57BL/6 mice for preparing BMDMs, but  12-month old L2-IL-1B mice for preparing 3D organoid culture in Matrigel?. Can the mice age have any effcet on the authors finding?
  3. Page 5. Similar to TLR2 mRNA observations, no TLR2 protein expression was detected in FLO-1 cells (Figure 1b). However, authors did not explain why the TLR2 mRNA expression level was very low in GO cells but TLR2 protein levels was high. Figure 1a did not include any statistical analysis, why?
  4. Page 9. In contrast, no TLR activation was observed in THP1-XBlue-CD14 cells incubated with CM from LPS-stimulated esophageal cell lines, which may be due to lower cellular responses to that ligand (Figure 1c-e). I think here is Figure 4c-e. Authors should discuss more this results. Which LPS of suppl Figure 1 was used in the experiments of figure 4?

Author Response

Reviewer 3 - Comments on revised manuscript

This is an interesting manuscript, as the authors identified a mechanism involved in TLR2-mediated inflammation in esophageal cells.  The authors find that the endogenous TLR2 ligand, HMGB1, is the factor secreted from EAC cells responsible for the observed TLR2-mediated effects on macrophages. Finally, authors suggest that blocking TLR2 signalling represents a mechanism to limit HMGB1 release, inflammatory cell infiltration and inflammation during EAC progression. These results can be important to increase our knowledge about molecular mechanisms of esophageal adenocarcinoma. However, some issues should be addressed.

  1. My main concern is that at this point the authors do not know whether or not HMBG1 represents the major TLR2 activating component secreted by EAC cells.

Response to Reviewer 3, Point 1: Thanks to reviewer #3 for their critical review of the manuscript. We acknowledge that HMGB1 may not be the only TLR2 activating component being secreted by EAC cells and have addressed this point in the discussion section (pg 16, highlighted in yellow). Figure 5 shows that TLR2-mediated HMGB1 secretion and upregulation occurs in EAC cells (which is inhibited when TLR2 is blocked) and Figure 6 shows that incubation of primary macrophages with recombinant HMGB1 mimics the pro-inflammatory TLR2-mediated effects of conditioned media from EAC cells – these results confirm that HMGB1 contributes to TLR2-mediated inflammation during EAC disease progression.

  1. Why the authors used 10-12 week-old C57BL/6 mice for preparing BMDMs, but 12-month old L2-IL-1B mice for preparing 3D organoid culture in Matrigel? Can the mice age have any effect on the authors finding?

Response to Reviewer 3, Point 2: We have used these mice for two different experimental systems. We used wild-type C57BL/6 mice for macrophage isolation, as standard protocol is to use healthy mice aged 10-12 weeks for the isolation of primary bone-marrow derived macrophages (BMDM). In contrast, we used 12 month old transgenic L2-IL-1B mice to generate organoid models of Barrett’s esophagus. It takes 12 months for the L2-IL-1B mice to develop well established metaplasia (Barrett’s) and initial signs of dysplasia. As the organoid and macrophage experiments were not being used in the same experimental system, we did not consider it necessary to isolate BMDM from similarly aged (12  month old) mice.

  1. Page 5. Similar to TLR2 mRNA observations, no TLR2 protein expression was detected in FLO-1 cells (Figure 1b). However, authors did not explain why the TLR2 mRNA expression level was very low in GO cells but TLR2 protein levels was high. Figure 1a did not include any statistical analysis, why?

Response to Reviewer 3, Point 3: Figure 1a and 1b show that, while low basal TLR2 expression was observed at both mRNA and protein levels, TLR2 stimulation induced robust upregulation of its expression in GO, SK-GT 4 and OE33 cells. We have explained this more clearly in the results section (page 5 – highlighted in blue) and have replaced the TLR2 and actin blots in Figure 1b with blots from a replicate experiment, which compliment mRNA data shown in Figure 1a. We did not originally include statistical analysis for mRNA expression in Figure 1a as its purpose is to show basal expression in individual cell lines. However, we have now included statistics to reflect that basal expression of TLR2 mRNA in GO, SK-GT 4 and OE33 is significant, relative to HEK293 cells, which do not express TLR2 (highlighted in Figure 1a legend).

  1. Page 9. In contrast, no TLR activation was observed in THP1-XBlue-CD14 cells incubated with CM from LPS-stimulated esophageal cell lines, which may be due to lower cellular responses to that ligand (Figure 1c-e). I think here is Figure 4c-e. Authors should discuss more this results. Which LPS of suppl Figure 1 was used in the experiments of figure 4?

Response to Reviewer 3, Point 4: We have discussed this result more clearly in the revised manuscript (pg 9, highlighted in blue), ‘In contrast, no THP1-XBlue-CD14 cell responses were observed following incubation with CM from esophageal cell lines stimulated with the TLR4 agonist LPS, providing further confirmation that the secretion of pro-inflammatory factors from diseased esophageal cells is TLR2 dependent (Figure 4 a-c).

Unpurified LPS (Sigma) was used to generate CM from LPS-stimulated EAC cells – we have also included this information in the materials and methods (section 2.4, pg 3).

This manuscript is a resubmission of an earlier submission. The following is a list of the peer review reports and author responses from that submission.

Round 1

Reviewer 1 Report

Very nice work and scientifically sound. 

Could you hypothesise if there are any other factors in the CM that might cause activation of macrophages? You have shown that HMGB1 is increased in CM via TLR2 but this is possibly/probably not the only factor that is released into the CM that is causing macrophage activation and cytokine release. 

How might future work identify whether this is the main factor causing this activation, or identify what factor(s) is/are responsible?

Author Response

Response to Reviewer 1, Point 1: We have addressed this point in the discussion section (highlighted text): ‘Future studies will determine whether, in addition to HMGB1, esophageal tumour cells also secrete factors including extracellular matrix components (eg. versican, biglycan, heparan sulphate) and heat shock proteins, which have been previously shown to be secreted from tumour cells to activate macrophages via TLR2-induced inflammation. Future studies designed to immunodeplete HMBG1 from EAC CM will confirm whether HMGB1 represents the major TLR2 activating component secreted by EAC cells’.

Reviewer 2 Report

The role of inflammation in the development of eosphageal adeno carcinoma (EAC) and Barrett Oesophagus (BO) is interesting topic with previous studies showing the involvement of Toll like receptors in tumourigenesis. In this submission Flis et al study the activation of TLR2 in three EAC and Barrett cell lines and provide preliminary evidence for the production of an endogenous activator by the tumour cells, the chromatin protein HMGB1.

One general problem is an over reliance on Western blot analysis and in particular inappropriate quantitation. W. blot is at best semi-quantitative and densitometry can give very misleading results due to the small linear range. While this approach can produce useful results when the blot is of high quality and the exposure is carefully controlled some of the blots in this study are of poor quality and in my view unsuitable for meaningful quantitation.

In Fig. 1 the expression and signalling of TLR2 in the EAC cell lines is analysed. All 3 cell lines had TLR2 RNA and ligands induced IL-8 and IL-6 production. This conclusion was confirmed in Fig 2 where an antagonistic anti-TLR2 Mab inhibited ligand induced cytokine. Next they used a mouse model of BO to generate organoids that are also responsive to TLR2 synthetic ligand PAM2CSK4 and produce MIP-1 and CXCL1.

The final experiments are less convincing. In Fig. 4  the EAC cell lines were untreated or treated with anti-TLR2 Ab then with TLR2 ligand for 4h. The cells were then washed and new media added for 24h. This conditioned media (CM) was used to assay for TLR2 activity in a THP-1 reporter cell line. I find the logic of this experiment difficult to follow. How is TLR2 activation sustained after the medium containing ligand is removed – surely any residual activity would dissipate rapidly? What is the half life of the ligand in culture and how easily is it washed out? Nevertheless, on the face of it they do see some activation by CM but this experiment could do with a control to rule out carry over of ligand. Next they ask whether HMGB1, well characterised as an endogenous activator of TLR4, could be secreted to the CM and responsible for TLR2 activity. They treated one of the EAC cell lines with PAM2CSK4 and then probed ‘supernatant’ and ‘cytoplasm’ for HMGB1 (Fig 5B). This is not equivalent to the experiment in Fig. 4 because the cells are treated continuously with ligand. The fractionation of the cells is not described and positive controls are needed for a cytoplasmic marker and also for any cell lysis releasing cytoplasm into the media supernatant. The quality of the blot presented is very poor and needs to be repeated with the controls – it is not publishable as presented. In Fig 6a mouse BMDMs are stimulated with recombinant HMGB1 and upregulation of caspase 11 and IL1beta measured. It is not stated where the rHMGB comes from – this is important because if it was expressed in E. coli it is almost certainly contaminated with TLR activators. The final experiments look at cytokine and inflammasome production in BMDM treated with CM. The quality of the blots in Fig 6 is also poor.

In conclusion in my opinion some additional experiments are needed for this to be publishable.

Author Response

  1. One general problem is an over reliance on Western blot analysis and in particular inappropriate quantitation. W. blot is at best semi-quantitative and densitometry can give very misleading results due to the small linear range. While this approach can produce useful results when the blot is of high quality and the exposure is carefully controlled some of the blots in this study are of poor quality and in my view unsuitable for meaningful quantitation.

Response to Reviewer 2, Point 1: We thank Reviewer #2 for their comments and agree that some of the Western blot quality needed to be improved – we have replaced the revised manuscript figures with better quality images for blots, particularly in Figures 5 and 6.

  1. In Fig. 1 the expression and signalling of TLR2 in the EAC cell lines is analysed. All 3 cell lines had TLR2 RNA and ligands induced IL-8 and IL-6 production. This conclusion was confirmed in Fig 2 where an antagonistic anti-TLR2 Mab inhibited ligand induced cytokine. Next they used a mouse model of BO to generate organoids that are also responsive to TLR2 synthetic ligand PAM2CSK4 and produce MIP-1 and CXCL1. The final experiments are less convincing. In Fig. 4 the EAC cell lines were untreated or treated with anti-TLR2 Ab then with TLR2 ligand for 4h. The cells were then washed and new media added for 24h. This conditioned media (CM) was used to assay for TLR2 activity in a THP-1 reporter cell line. I find the logic of this experiment difficult to follow. How is TLR2 activation sustained after the medium containing ligand is removed – surely any residual activity would dissipate rapidly? What is the half-life of the ligand in culture and how easily is it washed out? Nevertheless, on the face of it they do see some activation by CM but this experiment could do with a control to rule out carry over of ligand.

Response to Reviewer 2, Point 2: The experimental hypothesis has now been explained more clearly in the revised manuscript (Results section, Figure 4). In this Figure, we stimulated cells and washed off the stimulating ligand so that we could determine whether the EAC cells were secreting TLR2 activating ligands. Following TLR2 activation, HMGB1 is getting upregulated and then translocated to cytoplasm and released from the cells over time.  We have included time-courses as control experiments for each EAC cell line, to show that media from cells where the ligand has been removed by washing contains only basal levels of TLR2 activating ligands at 1 min, but these levels increase over time, as endogenous TLR2 ligands are secreted from the EAC cells and into the media (Figure 4 d-f).

  1. Next, they ask whether HMGB1, well characterised as an endogenous activator of TLR4, could be secreted to the CM and responsible for TLR2 activity. They treated one of the EAC cell lines with PAM2CSK4 and then probed ‘supernatant’ and ‘cytoplasm’ for HMGB1 (Fig 5B). This is not equivalent to the experiment in Fig. 4 because the cells are treated continuously with ligand. The fractionation of the cells is not described and positive controls are needed for a cytoplasmic marker and also for any cell lysis releasing cytoplasm into the media supernatant. The quality of the blot presented is very poor and needs to be repeated with the controls – it is not publishable as presented.

Response to Reviewer 2, point 3: We continuously stimulated cells in Figure 5 (rather than washing them after 4h stimulation) as it is difficult to detect secreted HMGB1 in supernatants from cells, and we wanted to maximise our ability to detect secreted HMGB1. The methodology for the fractionation of cells and isolation of cytosolic fractions is now included in the Materials and Methods section (please see tracked changes in revised manuscript). In Figure 5b, we have also repeated the fractionation experiments and included better quality blots. We have also included whole cell lysates as positive controls, probed for Lamin B1 as a nuclear marker and b-actin as a cytoplasmic marker. Stimulation with TLR agonists does not generally cause cell lysis, however we have carried out LDH assays on supernatants to confirm that cell lysis is not occurring under these conditions (LDH results can be included in a supplementary figure if Reviewer 2 thinks that it is necessary?). 

  1. In Fig 6a mouse BMDMs are stimulated with recombinant HMGB1 and upregulation of caspase 11 and IL1beta measured. It is not stated where the rHMGB comes from – this is important because if it was expressed in E. coli it is almost certainly contaminated with TLR activators. The final experiments look at cytokine and inflammasome production in BMDM treated with CM. The quality of the blots in Fig 6 is also poor. In conclusion in my opinion some additional experiments are needed for this to be publishable.

Response to Reviewer 2, point 4: The source of recombinant HMGB1 has been included in the revised Methods section, it is derived from human HEK 293 cells. The quality of blots in Figure 6 has been improved.

Reviewer 3 Report

The authors well-presented the interesting and important findins of their study. It was a pleasure to read and review this paper.

However, some minor points could been improved:

1) Figure 2 does not correspond with the text. In the paragraph 3.2. you wrote that GO cells are figures 2a and 2b, while SK-GT4 cells are presented in Figures 2c and 2d. However, in the figure 2, GO cells are represented by a and b, while SK-GT4 cells are presented in c and d.

2) Suplementary figure 1. Please change „Listeria Monocytogenes” into „Listeria monocytogenes”; „Salmonella Typhimurium” into „Salmonella typhimurium”; „Lactobacillus Rhamnosus” into „Lactobacillus rhamnosus”; All bacteria names should be written in italics; the first part is by capital letter and the second by small letter . Additionally, please change the first E. coli” into „Escherichia coli”. The abbreviation E.coli can be misleading because it can be Entamoeba coli (commensal parasite) or Escherichia coli (bacteria)

Author Response

Response to Reviewer 3, Point 1: Thanks to the reviewer for spotting this error, the Figure 2 graph labels have been corrected in the revised manuscript.

Response to Reviewer 3, point 2: The labelling in revised Supplementary Figure 1 has been corrected accordingly.